# Decomposition of income-related inequality in health check-ups services participation among elderly individuals across the 2008 financial crisis in Taiwan

Chiao-Lee Chu[1]*, Nozuko Lawana[2]

1 Department of Long Term Care, National Quemoy University, Kimmen County, Taiwan, 2 Social Policy, Knowledge Mobilization and Impact Assessment (SoKIA), Human Science Research Council (HSRC), Pretoria, South Africa

* chiaoleechu@gmail.com

**Data Availability Statement:** The data that support the findings of this study are available from the National Health Research Institutes (NHRI), Taiwan (website: http://nhis.nhri.org.tw/2009nhis.html;

## Abstract

Encouraging citizens to use health checkup services is a health promotion strategy. In nations with aging populations, ensuring equitable use of health check-ups by senior citizens is a public health concern. The objective of this research was to quantify income-related inequality and its effect on the use of health checkup services in Taiwan during the 2007–2008 global financial crisis. We used the 2005 and 2009 datasets of the Taiwan National Health Interview Surveys to assess how income-related inequality influenced health check-up use among older adults in Taiwan during the 2007–2008 financial crisis. Corrected concentration indices (CCIs) were calculated and decomposed to determine the influences of explanatory variables. The dependent variable was whether participants had used free senior health check-ups in the past year, and the determinant factors were health behavior, health situation, socioeconomic and demographic factors, and area health care resources accessibility factors. The study assessed 2,460 older adults from the 2005 dataset and 2,514 such individuals from the 2009 dataset. The utilization of health check-ups increased from 21.6% in 2005 to 34.0% in 2009. Income-related inequality in the use of health check-up services was generally tilted toward the higher income individuals among both women and men in 2005 and 2009, and income-related inequality decreased among women group and increased among men group with non significantly from 2005 to 2009 (women: CCI decreased from.0738 in 2005 to.0658 in 2009; men: CCI increased from.1068 in 2005 to.1256 in 2009). We analyzed the effect of explanatory factors on men's and women's intention to use health check-ups by using a probit model. After controlling for other factors, we determined that income significantly influenced women's health check-up service use in 2005 and men's in 2005 and 2009. Positive health behavior significantly increased health check-up services use among men and women group after the financial crisis, and negative health behavior significantly reduced health check-ups use among men across financial crisis. The 2008 global financial crisis strengthened the effect on health check-ups use of income-related inequality of elderly men, especially in older adults with negative health behaviors. Elderly men with negative health behaviors tended to contribute more

http://nhis.nhri.org.tw/2005nhis.html). However, restrictions apply to the availability of these data, which were used under license for the current study, and so are not publicly available. Requests for data can be sent as a formal proposal to the NHRI (http://nhis.nhri.org.tw) or contact the person in charge of NHIS data set at nhisis@nhri.edu.tw. These data are third party. The authors confirm that others would be able to access these data in the same manner as the authors, and the authors did not have any special access privileges that others would not have.

**Funding:** The author(s) received no specific funding for this work.

**Competing interests:** The authors have declared that no competing interests exist.

income-related inequality in use health check-up services after the financial crisis. Health promotion initiatives should focus their efforts on elderly men with negative health behaviors.

## Introduction

Promoting the health of elderly individuals is crucial in countries with aging populations. Offering health check-up services to elderly individuals is a key health promotion strategy; health check-up services aims for health promotion and disease prevention and typically entails reviewing a patient's history and making a comprehensive physical examination [1]. Freedom from chronic diseases is an indicator in the framework of healthy and active aging [2] proposed by the World Health Organization [3]. Older individuals with better health benefit from continued social activity and roles in the labor market [4, 5]. As a means of addressing its aging population Taiwan provides a free health check-up services program through the National Health Insurance (NHI) program.

NHI covers not only comprehensive medical care but also preventive care services. For all residents aged 65 and over, NHI provides free periodic health checkups (free adult health check-up services) that entail a physical examination, health counseling, and routine blood and urine examinations. The physical examination includes examination of height, weight, hearing, vision, oral health, and blood pressure; health counseling includes nutrition and diet counseling, encouragement to quit smoking and betel nut chewing, recommendations for age-appropriate physical activity and exercise, accident prevention strategies, and psychological adjustment counseling [6]. Health check-up services can keep senior citizens informed of their physical status, and doctors can provide suggestions regarding the diet and exercise needs of older adults, according to the results of the physical examination.

Most studies have concluded that health check-up services can promote health; such services can reduce inpatient and outpatient service use and expenditures [7–10], and the probability of curative care [11]. Moreover, the use of annual general health examinations or specific screening services can reduce mortality among middle-aged and older citizens [12, 13]. Studies have documented the relationships between the use of health check-up services and sociodemographic factors [14–18]. However, neither to what extent income inequality influences the use of such services nor how the financial crisis influenced the unequal use of such services has been evaluated. The financial meltdown of 2007 and 2008 was the most critical economic downturn in the past two decades, and it began with the subprime mortgage crisis in the United States; the effects of that crisis were far reaching, rattling the entire global economy. Taiwan was most affected in the second half of 2008 [19]. The major contribution of this study is the policy inferences that can be drawn from clarifying how the financial crisis affected the use of health check-ups by older adults; the lessons taught by this study can lead to more equitable utilization of such services.

This study estimated the effect of income-related inequality on the use of free senior health check-ups among elderly Taiwanese individuals in 2005 and 2009 through the use of concentration indices. Additionally, inequality was decomposed to determine the changes in the contribution of each explanatory factor over time.

## Materials and methods

### Database

Using a cross-sectional study design, we analyzed two cross-sectional survey datasets, one from 2005 and one from 2009, comprising responses to the Taiwan National Health Interview Survey (NHIS), issued by the Taiwan Health Promotion Administration of the Ministry of Health and Welfare. NHIS data are representative of the population of Taiwan [20, 21]. A multistage systematic stratified sampling design with probability proportional to size sampling (PPS) was employed. According to the geographical location, population density and degree of urbanization of the townships/districts, the first step of sampling scheme was to divide 358 townships/districts of Taiwan into 53 strata in 2005 and 48 strata in 2009. The selection probability of township/district was with PPS. For each selected township/district, lins (the smallest administrative unit) were selected with PPS. In each selected line, respondents were selected with PPS. The total sampling rate is 1.34 ‰ in 2005 and 1.32 ‰ in 2009.

Given the purpose of this study, we analyzed samples of survey respondents aged ≥65 years and selected the personal characteristics, health conditions, health behaviors, preventive care utilization and socioeconomic factors as this study analysis variables.

### Inequality measurement and decomposition

We used the concentration index (CI) [22, 23] to measure and decompose income-related inequality associated with the use of health check-ups. The CI can be calculated according to the following formula:

$$CI = \frac{2}{\bar{y}} cov(y_i, R_i)$$

where $\bar{y}$ is the mean value of the health status proxy (i.e., the health check-up services utilization), $y_i$ is the health status of $i$ th individual, and $R_i$ is a cumulative percentage that each individual represents over the total population after population has been ranked by income. When the health status is concentrated in individuals with relative poor (rich) income, the CI will show a negative (positive) value. CI values range from −1 to 1. A CI of 0 means that the health status is distributed equally among income brackets.

To compare ill health status between individuals and groups, Erreygers proposed using the corrected CI to handle cases of dichotomous variables (i.e., those that can be only 0 or 1) [24]. Corrected CI (CCI) is calculated as follows:

$$CCI = \frac{4\bar{y}}{y^{max} - y^{min}} CI$$

where $y^{min}$ and $y^{max}$ are the highest and lowest values of the variable.

Decomposition of the CI is possible using regression techniques; it can thus be used to quantify the contributions of various factors to income-related inequality in the use of health check-up services [25]. We can calculate the contribution of determinants to health check-ups inequality by using Wagstaff decomposition method [26], is calculated as follows:

$$CCI = 4 \sum_k (\beta_k^m \bar{x}_k) CI_k + GCI_\varepsilon$$

where $\bar{x}_k$ represents the mean value of explanatory variables, $\beta_k^m$ represents the partial effects evaluated using sample means, $CI_k$ represents the CI of determinant $X_k$, and $GCI_\varepsilon$ represents the generalized CI for the error term.

### Definition of variables

**Inequality CI.** A continuous variable is required to measure inequality using decomposition of the CI to rank members of the population socioeconomic status. The NHIS datasets for 2005 and 2009 include monthly income as a category with 10 response intervals. Consequently, generating a continuous ranking variable from these income categories was necessary. A continuous income variable was calculated using the demographic data of survey respondents and interval regression; covariates were education level, age, age squared, mean household income for region of residence, and prior working status. We used this method to separately compute the incomes of older women and men in 2005 and 2009. The monthly income of older adults was set at greater than NT\$1, and equivalent income was calculated using the modified equivalence scale of the Organization for Economic Co-operation and Development as well as consumer price indices from the Taiwan Executive Yuan's Directorate General of Budget, Accounting, and Statistics. Equivalent income was scaled to 2009 New Taiwan Dollars.

**Dependent variable.** The response to the question on the NHIS questionnaire "Have you used any health check-up provided by NHI in [the] past one year?" was used as the dependent variable. Participants responded "yes" or "no" to this question, thus making it a dichotomous, variable.

**Explanatory variables.** Health checkups is a demand for health and lead to further demand for preventive or medical care when necessary [27–29]. Use of health check-up services depends on user's characters and various other factors [27], that includes demographic, socioeconomic, personal health conditions, and the health care resources accessibility factors [18, 30, 31]. We also included the health behavior variable (life style) as an explanatory variable because it was demonstrated to be associated with health check-ups utilization [31]. This study selected the following explanatory variables for assessment: self-rated health, chronic disease status, and mobility difficulties (individual's health factors), age, marital status, and educational level (proxies for demographic factors), the log of predicted income (income variable), and the number of physicians per 10,000 persons (as a proxy of health care resources accessibility factor). Health behavior factors were the following: current smoking, current drinking, current betel nut chewing (all negative health behaviors), and engagement in exercise in the past 2 weeks (a positive health behavior).

### Statistical analysis

StataSE 13 was used to conduct statistical analyses, and the conindex command was employed to calculate CIs [32]; individual weighted values were given to data points so that the samples represented the general population.

Data analysis employed a probit model. The factors that influence health or health/preventive care utilization differ between men and women [33–38]. We divided respondents into two groups: men and women, and these two groups were separately analyzed in 2005 and 2009.

### Ethics statement

We utilized the Taiwan NHIS which was available for academic research. This study adhered to strict confidentiality guidelines that were in accordance with the regulations regarding personal electronic data protection. Because our research is secondary data analysis, this project meets the criteria for exemption from further review by the Human Research Ethics Committee at National Cheng Kung University (HREC No. 108–221). As the data files were de-identified, written consent was not required.

## Results

This study analyzed 2,460 survey respondents from 2005 and 2,514 from 2009. Table 1 lists all the variables related to health check-ups use in the past year, as determined by assessing the survey datasets (2005 and 2009), and their distribution in the samples. Use of health check-up services increased from 2005 (21.6%) to 2009 (34.0%). The demographics of users of health check-up services are presented in Table 1.

The income CIs for the use of health check-up services are presented in Table 2. Those with relatively high incomes used health check-up services more than other respondents. This result was statistically significant for both 2005 and 2009. For women, income-related inequality went from concentrated among those with relatively high incomes (p value = 0.0682) in 2005 to less pro-rich concentrated with significantly (p value = 0.0457) in 2009. However, a statistically significant change was not evident between study periods. Among men, income-related inequality significantly favored high-income respondents, but no statistically significant change was evident between study periods. The CI of older women decreased from.0738 in 2005 to.0658 in 2009, thus indicating more health check-ups use by those with higher incomes in both years. The CI exhibited a increasing trend for men, from.1068 in 2005 to.1256 in 2009. The CIs reveal that inequality in health check-ups utilization was higher among men than women.

Tables 3 and 4 present the probit and inequality decomposition results for women and men, respectively. Table 3 presents the factor contributions to the inequality in the use of health check-up services in 2005 and 2009 among elderly women. The four columns present the estimations of the partial effects from the probit model, corrected CI of each regressor, absolute contribution of each explanatory variable, and the CI percentage contribution, respectively, for each year. Here, the partial CI for every determinant is presented in the second column of the tables. A positive or negative CI indicates that the distribution of the variable is concentrated among relatively high- or low-income individuals, respectively. For example, for women in 2009, an age of 65 to 74 years, a junior high school or higher level of education, and exercise during the previous 2 weeks had positive CIs. Thus, the concentrations of these factors were high among high-income older women. For each factor, the absolute contributions to income-related inequality are presented in the third column. The effect attributable to income of each variable on the distribution of health check-up service use is the absolute contribution. The percentage contribution calculates from dividing the absolute contribution by the overall income-related inequality is reported in the fourth column. For an explanatory variable, a positive absolute contribution indicates that if inequality related to health check-up service use were determined by that variable only, inequality would favor high-income individuals. That is, a negative or positive absolute contribution indicates that the inequality in health check-ups use would increase or decrease, respectively, if that variable were distributed equally across the wealth distribution.

Table 3 presents the partial effects that indicate that women with relatively high income and who reside in locations with a higher physician–population ratio were significantly more likely to use health check-up services in 2005. In 2009, women aged 65–74 years who had a higher educational level, exercised in the previous 2 weeks, and self-reported poor health were used significantly more health check-up services.

Positive (negative) CCIs indicate that the distributions of explanatory variables were tilted toward those with a relatively high (low) income (Table 3). Inequality in health check-ups participation by the direct effect of income in both periods, it's from 93.36% in 2005 to -7.27% in 2009, and the contribution of income strongly decreased from.0689 in 2005 to -0.0048 in 2009.

**Table 1. Study sample characteristics by health check-ups (health examination) use and year, *n* (%).**

| Variables | 2005 | | 2009 | |
|---|---|---|---|---|
| **Dependent variable** | yes | no | yes | no |
| Preventive services utilization | 531 (21.6) | 1929 (78.4) | 855 (34.0) | 1659 (66.0) |
| **Independent variable** | | | | |
| Demographic | | | | |
| Gender | | | | |
| Female | 256 (20.8) | 971 (79.2) | 473 (33.4) | 943 (66.6) |
| Male | 275 (22.3) | 958 (77.7) | 382 (34.8) | 716 (65.2) |
| Marital status | | | | |
| Married | 359 (22.5) | 1237 (77.5) | 598 (37.1) | 1013 (62.9) |
| Unmarried | 172 (19.9) | 692 (80.1) | 257 (28.5) | 646 (71.5) |
| Age group | | | | |
| 65–74 | 304 (19.8) | 1230 (80.2) | 515 (34.7) | 969 (65.3) |
| 75+ | 227 (24.5) | 699 (75.5) | 340 (30.0) | 690 (67.0) |
| Education Level | | | | |
| informal education or illiterate | 189 (19.6) | 776 (80.4) | 217 (28.4) | 547 (71.6) |
| elementary education | 208 (21.1) | 775 (78.9) | 402 (34.2) | 772 (65.8) |
| junior high school or above | 134 (26.2) | 378 (73.8) | 236 (41.0) | 340 (59.0) |
| Number of individuals living together | Mean (s.e) | Mean (s.e) | Mean (s.e) | Mean (s.e) |
| | 2.88 (2.58) | 3.22 (2.97) | 2.68 (2.31) | 2.78 (2.47) |
| Bad health behavior | | | | |
| Drinking | | | | |
| Yes | 103 (20.4) | 401 (79.6) | 158 (34.0) | 307 (66.0) |
| No | 428 (21.9) | 1528 (78.1) | 697 (34.0) | 1352 (66.0) |
| Smoking | | | | |
| Yes | 114 (17.4) | 540 (82.6) | 83 (26.2) | 234 (73.8) |
| No | 417 (23.1) | 1389 (76.9) | 772 (35.1) | 1425 (64.9) |
| Betel nut chewing | | | | |
| Yes | 12 (12.0) | 88 (88.0) | 70 (27.4) | 185 (72.6) |
| No | 519 (22.0) | 1841 (78.0) | 785 (34.8) | 1474 (65.2) |
| Good health behavior | | | | |
| Whether or not do exercise in past two weeks | | | | |
| not doing any exercise in past two weeks | 195 (18.7) | 847 (81.3) | 352 (29.1) | 857 (70.9) |
| doing any exercise in past two weeks | 336 (23.7) | 1082 (76.3) | 503 (38.5) | 802 (61.5) |
| Health condition | | | | |
| Self-reposted health | | | | |
| Bad | 337 (21.2) | 1250 (78.8) | 549 (33.9) | 1070 (66.1) |
| Fair | 115 (21.9) | 409 (78.1) | 196 (33.9) | 382 (66.1) |
| Good | 79 (22.6) | 270 (77.4) | 110 (34.7) | 207 (65.3) |
| Chronic disease | | | | |
| with chronic disease | 296 (24.8) | 897 (75.2) | 519 (36.4) | 907 (63.6) |
| without chronic disease | 235 (18.5) | 1032 (81.5) | 336 (30.9) | 752 (69.1) |
| Mobility | | | | |
| with any mobile difficulty | 268 (20.6) | 1032 (79.4) | 421 (33.7) | 829 (66.3) |
| without any mobile difficulty | 263 (22.7) | 897 (77.3) | 434 (34.3) | 830 (65.7) |
| Area health care resource | | | | |
| Physician no. per 10000 populations in area[a] | | | | |
| 1: for under 70 physicians | 225 (23.2) | 744 (76.8) | 97 (34.6) | 183 (65.4) |

(*Continued*)

**Table 1.** (Continued)

| Variables | 2005 | | 2009 | |
|---|---|---|---|---|
| 2: for 71–90 physicians | 152 (18.1) | 686 (81.9) | 417 (34.0) | 809 (66.0) |
| 3: for 91 and above physicians | 154 (23.6) | 500 (76.4) | 341 (33.8) | 667 (66.2) |
| Personal income | | | | |
| Lpinco [b] | Mean (s.e) | | Mean (s.e) | |
| | 8.43 (2.29) | 7.78 (3.00) | 8.69 (2.12) | 8.41 (2.39) |

Notes:

[a]Ratio: 1 if 70, 2 if 71–90, and 3 if 91 or more physicians per 10,000 persons.

[b]log of equivalent income (2009 new Taiwan dollars).

Preventive care utilization: use of preventive care in the past one year.

s.e.: standard error.

This result indicates that the proportion of total inequality attributable to income significantly decreased in women group. (Table 3).

Among men group, the income variable (Lpinco) significantly influenced the use of health check-up services in 2005 and 2009 (Table 4). Moreover, 65–74 year old men used health check-up services significantly less in 2005 (Table 4). Older men who had exercised in the past 2 weeks or had any chronic disease were significantly more prone to use health check-up services in 2005. Married men used health check-up services significantly more in 2009. Additionally, health behavior, as expected, influenced men's use of health check-ups. Positive health behavior (exercise in previous 2 weeks) had a statistically significant positive effect, and negative health behaviors (drinking, smoking and chewing betel nut) had an increasing percentage contribution effect on health check-up services use from 4.83% in 2005 to 11.38% in 2009.

The signs of the CCIs shown in Table 4 indicate whether the distributions of the explanatory variables favored individuals with high (positive) or low (negative) incomes. Inequality in health check-ups participation by the direct effect of income in both periods, it increased from .0782 in 2005 to .0826 in 2009. But the percentage contribution of income decreased to 65.76% in 2009 from 73.22%. This result indicates that the proportion of total inequality attributable to income slightly decreased in men group.

## Discussion

Older adults with relatively high incomes used significantly more health check-up services in 2005 and 2009 in Taiwan, demonstrating income-related inequality. However, the trend of inequality decreased in women group, and increase increased in men group across study periods, both of the change in inequality from 2005 to 2009 was not statistically significant.

**Table 2. CIs of unequal health check-ups use for 2005 and 2009.**

| | 2005 | | 2009 | | difference | |
|---|---|---|---|---|---|---|
| | Index value (s.e.) | p vaule | Index value (s.e.) | p vaule | Diff (s.e.) | p value |
| Women | 0.0738 (0.0405) | 0.0682 | 0.0660 (0.0328) | 0.0448 | - 0.0078 (0.0521) | 0.8806 |
| Men | 0.1068 (0.0401) | 0.0078 | 0.1256 (0.0374) | 0.0008 | 0.0189 (0.0548) | 0.7306 |

CIs: concentration indexs.

s.e.: standard error.

Diff: difference between 2005 and 2009.

**Table 3. Variable contributions to unequal use of health check-ups in 2005 and 2009 from probit regression, women.**

| variable | 2005 (n = 1227) | | | | 2009 (n = 1416) | | | |
|---|---|---|---|---|---|---|---|---|
| | coefficient | Corrected CI | contribution | % contribution | coefficient | Corrected CI | contribution | % contribution |
| Lpinco | 0.0464 ** | 0.1786 | 0.0689 | 93.36 | - 0.0041 | 0.1025 | -0.0048 | -7.27 |
| Demographic | | | | | | | | |
| Age group (ref = 75+) | | | | | | | | |
| 65–74 | - 0.0554 | 0.0030 | -0.0001 | -0.14 | 0.2332 * | 0.0767 | 0.0152 | 23.03 |
| Marital status (ref = unmarried) | | | | | | | | |
| Married | 0.1663 | - 0.0071 | -0.0007 | -0.95 | 0.1737 | 0.0101 | 0.0012 | 1.82 |
| Education Level (Ref = informal education or illiterate) | | | | | | | | |
| elementary education | - 0.0481 | 0.1699 | -0.0030 | -4.07 | 0.1255 | 0.0161 | 0.0011 | 1.67 |
| junior high school or above | 0.0596 | 0.3062 | 0.0024 | 3.25 | 0.3512 * | 0.4959 | 0.0394 | 59.70 |
| Number of individuals living together | - 0.0187 | - 0.0236 | 0.0016 | 2.17 | - 0.0215 | - 0.0263 | 0.0022 | 3.33 |
| Bad health behavior | | | | | | | | |
| Drinking | | | | | | | | |
| Yes | - 0.2121 | 0.1129 | -0.0020 | -2.71 | - 0.0670 | 0.2630 | - 0.0023 | -3.48 |
| Smoking | | | | | | | | |
| Yes | - 0.2208 | - 0.1234 | 0.0010 | 1.36 | - 0.1352 | - 0.1599 | 0.0004 | 0.61 |
| Betel chewing | | | | | | | | |
| Yes | - 0.0950 | - 0.2985 | 0.0004 | 0.54 | - 0.1879 | - 0.1173 | 0.0005 | 0.76 |
| Good health behavior | | | | | | | | |
| Whether or not dosing exercise in past two weeks (Ref = not doing) | | | | | | | | |
| doing any exercise | - 0.0083 | 0.0506 | -0.0003 | -0.41 | 0.2181 * | 0.0388 | 0.0059 | 8.94 |
| Health condition | | | | | | | | |
| Self-reposted health (Ref = fair) | | | | | | | | |
| Good | - 0.2423 | 0.0420 | -0.0013 | -1.76 | 0.1781 | 0.1876 | 0.0059 | 8.94 |
| Bad | - 0.0279 | - 0.0291 | 0.0006 | 0.81 | 0.1862 | - 0.0409 | - 0.0068 | -10.30 |
| Chronic disease (Ref = without) | | | | | | | | |
| with chronic disease | 0.0263 | 0.0300 | 0.0005 | 0.68 | 0.1022 | - 0.0176 | - 0.0015 | - 2.27 |
| Mobility (Ref = without mobile difficulty) | | | | | | | | |
| with any mobile difficulty | - 0.0294 | -0.0343 | 0.0007 | 0.95 | - 0.1195 | - 0.0633 | 0.0059 | 8.94 |
| Area health care resource | | | | | | | | |
| Physician numbers per 10000 populations in area [a] (Ref = 2) | | | | | | | | |
| 1 | 0.2326 * | -0.0156 | - 0.0014 | -1.90 | - 0.1774 | 0.0419 | - 0.0017 | - 2.58 |
| 3 | 0.3352 ** | 0.1267 | 0.0141 | 19.11 | - 0.0472 | 0.0500 | - 0.0014 | - 2.12 |
| Residual | | | -0.0076 | -10.30 | | | 0.0068 | 10.30 |
| Total | | | 0.0738 | 100% | | | 0.0660 | 100.0 |

Health Check-ups: health check-ups use in the past one year.

ref: reference group.

Lpinco: log of equivalent income (2009 new Taiwan dollars).

[a]ratio: 1 if <70, 2 if 71–90, 3 if 91 or more physicians per 10,000 persons.

*Significant at 95% level ($p < .05$),

**Significant at 99% level ($p < .01$),

***Significant at 99.9% level ($p < .001$).

**Table 4. Variable contributions to unequal use of health check-ups in 2005 and 2009 from probit regression, men.**

| variable | 2005 (n = 1233) | | | | 2009 (n = 1098) | | | |
|---|---|---|---|---|---|---|---|---|
| | coefficient | Corrected CI | contribution | % contribution | coefficient | Corrected CI | contribution | % contribution |
| Lpinco | 0.0625 ** | 0.1371 | 0.0782 | 73.22 | 0.0611 * | 0.1172 | 0.0826 | 65.76 |
| Demographic | | | | | | | | |
| Age group (ref = 75+) | | | | | | | | |
| 65–74 | - 0.2455 * | -0.0397 | 0.0063 | 5.90 | - 0.2298 | - 0.0890 | 0.0159 | 12.66 |
| Marital status (Ref = unmarried) | | | | | | | | |
| Married | - 0.0699 | 0.0246 | - 0.0015 | - 1.40 | 0.3034 * | 0.0220 | 0.0070 | 5.57 |
| Education Level (ref: informal education or illiterate) | | | | | | | | |
| elementary education | 0.0112 | - 0.0937 | - 0.0005 | - 0.47 | 0.2187 | - 0.1901 | - 0.0269 | -21.42 |
| junior high school or above | - 0.0534 | 0.3363 | - 0.0065 | - 6.09 | 0.1722 | 0.2994 | 0.0280 | 22.29 |
| Number of individuals living together | 0.0062 | - 0.0878 | - 0.0019 | - 1.78 | 0.0148 | - 0.0389 | - 0.0021 | -1.67 |
| Bad health behavior | | | | | | | | |
| Drinking | | | | | | | | |
| Yes | - 0.0369 | 0.0518 | - 0.0007 | - 0.66 | - 0.0006 | 0.1134 | 0.0000 | 0.00 |
| Smoking | | | | | | | | |
| Yes | - 0.2090 * | - 0.0429 | 0.0048 | 4.49 | - 0.2213 | - 0.0192 | 0.0014 | 1.11 |
| Betel chewing | | | | | | | | |
| Yes | - 0.2979 | - 0.0930 | 0.0015 | 1.40 | - 0.2945 * | - 0.1689 | 0.0129 | 10.27 |
| Good health behavior | | | | | | | | |
| Do any exercise in past two weeks (ref: not) | | | | | | | | |
| doing any exercise | 0.2391 * | 0.0785 | 0.0129 | 12.08 | 0.3638 ** | 0.0290 | 0.0080 | 6.37 |
| Health condition | | | | | | | | |
| Self-reposted health (ref: fair) | | | | | | | | |
| Good | 0.2132 | 0.1638 | 0.0065 | 6.09 | - 0.1200 | 0.1860 | - 0.0045 | -3.58 |
| Bad | - 0.0077 | - 0.0624 | 0.0003 | 0.28 | - 0.0686 | - 0.0554 | 0.0029 | 2.31 |
| Chronic disease (Ref = without) | | | | | | | | |
| with chronic disease | 0.3753 *** | 0.0646 | 0.0118 | 11.05 | 0.1682 | - 0.0130 | - 0.0015 | -1.19 |
| Mobility (ref: without mobile difficulty) | | | | | | | | |
| with any mobile difficulty | - 0.1724 | - 0.0615 | 0.0048 | 4.49 | 0.0545 | - 0.0225 | - 0.0006 | -0.48 |
| Area health care resource | | | | | | | | |
| Physician numbers per 10000 populations in area [a] (Ref = 2) | | | | | | | | |
| 1 | 0.0948 | - 0.1043 | - 0.0038 | -3.56 | 0.0144 | - 0.0419 | - 0.0001 | -0.08 |
| 3 | 0.0625 | 0.1546 | 0.0032 | 3.00 | - 0.1506 | 0.1007 | - 0.0093 | -7.40 |
| Residual | | | - 0.0086 | -8.05 | | | 0.0119 | 9.47 |
| Total | | | 0.1068 | 100.0 | | | 0.1256 | 100.0 |

Health Check-ups: health check-ups use in the past one year.

ref: reference group.

Lpinco: log of equivalent income (2009 new Taiwan dollars).

[a]ratio: 1 if <70, 2 if 71–90, 3 if 91 or more physicians per 10,000 persons.

*Significant at 95% level ($p < .05$),

**Significant at 99% level ($p < .01$),

***Significant at 99.9% level ($p < .001$).

Therefore, free adult health check-up services can protect lower-income older women adults from unequal health check-ups utilization during a financial crisis, but cannot protect men group. Our probit regression results demonstrate that income is an important variable, which increases health check-up services utilization. Other studies have found similar results. For example, Liu et al. (2016) studied the unequal preventive care use in a population of China, that is, a relatively high household income was correlated with a high probability of using physical examination services [39]. The percentage CCI change trend of income decrease between study periods in both groups, women group had a larger CCI percentage decreased trend than men group. The free adult health check-up services may be can protect older adults but the financial crisis still has more impact effect on men group than on women group. Why financial crisis had a more effect on men group? The reason may be that the percentage of older men who own assets (54.3%) is higher than that of older women (35.6%) in 2005 in Taiwan [40, 41], and the 2007–2008 financial crisis has a direct impact on people with assets [42].

In terms of the age effect on health check-ups utilization, the opposite results are showed among female and male groups. The female participation rate in 2009 is similar to the results of other studies [28], that is, the 65–74 age group has a higher health checkups utilization rate than group aged 75 and over. This result indicates that the lifecycle effect may dominate the health-risk effect among female group after 2008 financial crisis. The lifecycle effect [43] and the health-risk effect [28, 44] mean that the use of preventive care decrease and increase with age, respectively [43]. However, the result of age effect of among men group is that group aged 75 and over has a higher health checkups utilization than 65–74 age group in 2005, also is similar to some studies in Taiwan [45, 46]. And the age effect of men group shows that the health-risk effect may dominate the lifecycle effect before the 2008 financial crisis. The possible explanation is that older male group has a gender advantage in our culture, the opinion of health-risk effect is not shown among female group. We suggest that future research can further focus on the lifecycle effect/health-risk effect on the use of preventive health services from gender difference perspective.

Among both men and women, health behavior variables significantly influenced health check-up services use in 2009, after the financial crisis. This result is as similar as previous study [47]. For women, the use of health check-up was positively related to exercise in past two weeks in 2009, and showed an increased inequality percentage contribution from -0.41% in 2005 to 8.94% in 2009. For men, the use of health check-up services was positively related to exercise in the previous 2 weeks and negatively to chewing betel nut. Betel nut chewing is more likely among low- income individuals and exhibited a conspicuously increased inequality percentage contribution from 1.40% in 2005 to 10.27% in 2009. The distribution of exercise was more evident among those with relatively high incomes, but it had a decreased inequality percentage contribution of 6.37% in 2009 from 12.08% in 2005. Few studies have focused on the relationship between positive and negative health behaviors and inequality in the use of health check-up services during financial crises. Recently, Trujillo-Aleman et al. identified inequality in the health behaviors of couples and single mothers. Their results indicated that the prevalence of sleeping less than 6 hours per day increased during the financial crisis for both couples and single mothers [48]. Assessing their results with ours suggests that global financial crises may increase poor health behaviors, which may reduce the utilization of health check-up services.

## Limitations

This study had a limitation. The variable of estimated income may have introduced bias. Had we additional data, such as personal wealth information, the explanatory power of income

would have been improved. Despite this limitation, our study yields valuable insight about the income-related unequal use of preventive care services among older adults in Taiwan across the 2007–2008 financial crisis.

## Conclusions

We conclude that the global financial crisis strengthened the effect on health check-ups use of income-related inequality of elderly men, especially in older adults with negative health behaviors. This study results show that the contribution percentage of income to the inequality of health check-ups utilization has decreased, but the contribution of health behaviors to the use of health check-ups has increased, after the economic crisis. In a national health insurance system with free health examinations, increasing senior health check-ups utilization would focus on groups with negative health behavior, especially across financial crisis.

The Taiwan NHI ensures free health check-up services for older individuals, but the distribution of such services is still affected by income-related inequality. Equitable distribution of health check-up service accords with the health promotion strategies proposed by World Health Organization for older adults. A key finding of the present study is that the contribution of negative health behavior among men to absolute inequality increased during the financial crisis. This study is a powerful reminder to Taiwan and other countries with aging populations that simply removing financial barriers to accessing health check-up services is not necessarily sufficient to compel older adults to take advantage of those services. Additionally, men with poor health behaviors tend to be vulnerable during such crises and should be targeted by outreach efforts to prevent an increase in inequality similar to that found in this study. At this time when the world is facing the economic recession caused by the covid-19 pandemic, this study maybe can provide the elderly health promotion strategies for countries with aging populations.

## Supporting information

**S1 Table. Correlation matrix of independent variables.**
(DOCX)

**S2 Table. Correlation matrix of independent variables, 2005.**
(DOCX)

**S3 Table. Correlation matrix of independent variables, 2009.**
(DOCX)

**S4 Table. Correlation matrix of independent variables, female, 2005.**
(DOCX)

**S5 Table. Correlation matrix of independent variables, female, 2009.**
(DOCX)

**S6 Table. Correlation matrix of independent variables, male, 2005.**
(DOCX)

**S7 Table. Correlation matrix of independent variables, male, 2009.**
(DOCX)

## Author Contributions

**Conceptualization:** Chiao-Lee Chu, Nozuko Lawana.

**Data curation:** Chiao-Lee Chu.

**Formal analysis:** Nozuko Lawana.

**Investigation:** Chiao-Lee Chu.

**Methodology:** Chiao-Lee Chu.

**Writing – original draft:** Chiao-Lee Chu, Nozuko Lawana.

**Writing – review & editing:** Chiao-Lee Chu.

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
