## [Decision Letter · Decision Letter 0]

18 Feb 2021

PONE-D-20-31185

Decomposition of Income-Related Inequality in Health Check-ups Services Participation Among Elderly Individuals Across the 2008 Financial Crisis in Taiwan

PLOS ONE

Dear Dr. Chu,

Thank you for submitting your manuscript to PLOS ONE. After careful consideration, we feel that it has merit but does not fully meet PLOS ONE’s publication criteria as it currently stands. Therefore, we invite you to submit a revised version of the manuscript that addresses the points raised during the review process.

We look forward to receiving your revised manuscript.

Kind regards,

Xi Pan

Academic Editor

PLOS ONE

2. Please specify in the methods section how demographic information was extracted and stratified for analysis.

Reviewers' comments:

Reviewer's Responses to Questions

**Comments to the Author**

1. Is the manuscript technically sound, and do the data support the conclusions?

Reviewer #1: Yes

Reviewer #2: Yes

2. Has the statistical analysis been performed appropriately and rigorously? 

Reviewer #1: I Don't Know

Reviewer #2: Yes

3. Have the authors made all data underlying the findings in their manuscript fully available?

Reviewer #1: Yes

Reviewer #2: No

4. Is the manuscript presented in an intelligible fashion and written in standard English?

Reviewer #1: Yes

Reviewer #2: No

5. Review Comments to the Author

Reviewer #1: The authors have addressed an important issue in this manuscript that can influence decision making and policy making in their country.

This manuscript needs minor language revision.

Simplification of the keywords may improve the internet search for other researchers.

Abstract:

- The description of the results in the abstract is vague to some extent, better description is already found in the results section of the body of the manuscript.

Reviewer #2: This manuscript assesses how income inequality and the 2008 financial crisis influenced the unequal use of health checkup services among the elderly in Taiwan. It uses combined data from 2005 and 2009 Taiwan National Health Interview Survey, targeting a subset of the respondents aged 65 and over. The measure of inequality used to assess unequal use in health checkup services is the concentration index (CI), which is decomposed using regression to quantify the effect of several variables such demography, socioeconomic, health and healthcare accessibility on income-related unequal use of health checkup services. Finally, it utilises a logistic regression model to determine the effect of those variables on health checkup services utilisation. An increase in health checkup services utilisation from 2005 to 2009 is seen, and several variables with differing effects on health checkup services utilisation are identified, and these are shown to operate differently between men and women. These results are exciting. They highlight an important point that although the 2008 financial crisis influenced income-related inequality of health checkup utilisation among the elderly in Taiwan, other factors play roles that can be positive or negative depending on age and gender. The data is of good quality, and I have a few suggestions and comments for potential improvement of the manuscript.

1. The manuscript states that a “multistage systematic stratified sampling design…” was used. A few lines clarifying precisely what that means with respect to the data used will indeed be helpful (i.e., give the number of sampling stages and details of the various sampling stages). Also, I assumed that the 2005 data and 2009 data are responses from the same set of individuals, but the presentation of the results and discussion indicates I may be wrong. Perhaps the authors may want to clarify this from the onset.

2. In the CI calculation, a variable called "health status" is defined which the reader is led to believe as meaning "health checkup services utilisation". However, further down in the methods, another variable called "health status" is defined in the section called "Explanatory variables" which I suspect may be referring to something entirely different. Further clarification on this may be needed.

3. The overall study attempts to pinpoint gender-specific patterns to health checkup services utilisation and, therefore, analyse men and women separately. However, the evidence on which this initial reasoning is based are studies conducted in other populations: Socias et al. 2016, Cameron et al. 2010, Vaidya et al. 2012 and Brunner-Ziegler et al. 2013, are based on Canadian, American and Austrian populations. The same effects may or may not apply to the Taiwan population, and one can only be sure if that is tested empirically. An initial analysis that puts males and females together could be performed to establish whether there is a significant gender difference in health checkup services utilisation. If the authors did this, then it should be acknowledged in the manuscript.

4. Since some of the health behaviour variables, both positive and negative, may correlate with education level, which could be a source of confounding in the model thus rending some of these variables insignificant. Perhaps the manuscript should acknowledge this and provide data on the correlation between these variables.

5. If data restrictions do not preclude it, I would suggest additional data on respondents’ household (number of individuals in the household, and also a measure of the support available to them at home) should be considered in the model.

6. Tables 3 and 4 are a bit hard to follow. I would suggest they are broken up into two tables each, with one table looking at demography and socioeconomic factors, and the other looking at behaviour and health.

7. The effect of income on health checkup services utilisation between males and females is an important result that warrants more discussion than provided in the manuscript. The authors should also provide references to support the statements that "men bear more of a family's financial burden", and "older men have more assets than older women in Taiwan".

8. The paper referenced (Cropper, 1997) in discussing the age effect of female health checkup utilisation does not provide sufficient support for the results observed in this manuscript. Cropper reasons that how investment in health (checkups, dietary supplements, etc.) changes over the course of life cannot be determined because of the uncertainty of death. Unlike human capital investment which is high when people are young and declines over time. Moreover, the opposite results obtained for men in both 2005 and 2009 prove the point of Cropper that changes in health investment cannot be determined with any certainty without first assuming the certainty of death. Therefore, I would suggest that the authors discuss those results further and provide relevant support from the literature.

9. Lee et al. (2017) did observe similarly that healthy lifestyles lead to an increase in the utilisation of preventive health services. However, there is a clear age difference between the data used in that study and this one. Perhaps acknowledge this?

Minor Suggestions

1. Some of the sentences in the manuscript can benefit from the proper use of commas.

2. The frequent use of semi-colons in the manuscript makes some sentences unclear or hard to follow. It will help the reader if the semi-colons were to be removed, and such sentences are broken up into two or more simpler sentences.

3. The exact p values should be quoted in the manuscript, instead of just stating “significant” or “marginally significant”.

4. In some parts of the results and discussion, the manuscript describes people aged 65 – 74 as younger adults and those 75 and over as older adults. I think this description is quite problematic because, for some readers, older adults mean people aged 65 and over. To avoid such confusion, I think the manuscript should quote the exact age group being referred to.

5. I think. Abstract and page 12 “The CI of older women increased from .0738 in 2005 to .0658 in 2009…” should this be “decreased”?

6. I think. The last but one sentence of the abstract “…elderly men with negative health behaviours tended to contribution more income-related inequality…” should this be “contribute”?

7. Pages 13 and 14. “The percentage contribution calculating from dividing the absolute contribution…” should this be “calculated”?

6. PLOS authors have the option to publish the peer review history of their article (what does this mean?). If published, this will include your full peer review and any attached files.

Reviewer #1: No

Reviewer #2: No

---

## [Editor Report · Decision Letter 1]

30 Apr 2021

PONE-D-20-31185R1

Decomposition of income-related inequality in health check-ups services participation among elderly individuals across the 2008 financial crisis in Taiwan

PLOS ONE

Dear Dr. Chu,

Thank you for submitting your manuscript to PLOS ONE. After careful consideration, we feel that it has merit but does not fully meet PLOS ONE’s publication criteria as it currently stands. Therefore, we invite you to submit a revised version of the manuscript that addresses the points raised during the review process.

We look forward to receiving your revised manuscript.

Kind regards,

Xi Pan

Academic Editor

PLOS ONE
---

## [Author Response · Author response to Decision Letter 1]

6 May 2021

Journal Requirements: 

1. Please review your reference list to ensure that it is complete and correct. 

Response:

We have checked reference list of this manuscript. We confirmed the reference list is complete and correct. 

2. If you have cited papers that have been retracted, please include the rationale for doing so in the manuscript text, or remove these references and replace them with relevant current references. 

Response:

We did not cite papers that have been retracted. 

3. Any changes to the reference list should be mentioned in the rebuttal letter that accompanies your revised manuscript. 

Response:

We did not change the reference list. 

We have edited the hyperlinks of the doi and PMID of each reference of this manuscript for meeting the publication criteria of PLOS one.

4. If you need to cite a retracted article, indicate the article’s retracted status in the References list and also include a citation and full reference for the retraction notice.

Response:

We did not cite any retracted articles.

5.

Response:

We did not find any reviewers’ comment and did not find attachment file. 

6. While revising your submission, please upload your figure files to the Preflight Analysis and Conversion Engine (PACE) digital diagnostic tool, https://pacev2.apexcovantage.com/. 

Response:

This study is not with any figures. We do not upload figure files.

We did our best to revise the reference list and hope that this revision can meet the publication requirements of PLOS one.

Thank you for your help and patience.

---

## [Editor Report · Decision Letter 2]

26 May 2021

Decomposition of income-related inequality in health check-ups services participation among elderly individuals across the 2008 financial crisis in Taiwan

PONE-D-20-31185R2

Dear Dr. Chu,

We’re pleased to inform you that your manuscript has been judged scientifically suitable for publication and will be formally accepted for publication once it meets all outstanding technical requirements.

Kind regards,

Xi Pan

Academic Editor

PLOS ONE
---

## [Editor Report · Acceptance letter]

2 Jun 2021

PONE-D-20-31185R2 

Decomposition of income-related inequality in health check-ups services participation among elderly individuals across the 2008 financial crisis in Taiwan 

Dear Dr. Chu:

I'm pleased to inform you that your manuscript has been deemed suitable for publication in PLOS ONE. Congratulations! Your manuscript is now with our production department. 

Kind regards, 

on behalf of

Dr. Xi Pan 

Academic Editor

PLOS ONE